# Emerging Roles of Salicylic Acid in Plant Saline Stress Tolerance

**DOI:** 10.3390/ijms24043388

**Published:** 2023-02-08

**Authors:** Wei Yang, Zhou Zhou, Zhaohui Chu

**Affiliations:** 1State Key Laboratory of Crop Biology, College of Life Sciences, Shandong Agricultural University, Taian 271018, China; 2State Key Laboratory of Hybrid Rice, College of Life Sciences, Wuhan University, Wuhan 430072, China

**Keywords:** salicylic acid, plant immunity, salinity stress, phytohormones biosynthesis, molecular mechanism

## Abstract

One of the most important phytohormones is salicylic acid (SA), which is essential for the regulation of plant growth, development, ripening, and defense responses. The role of SA in plant–pathogen interactions has attracted a lot of attention. Aside from defense responses, SA is also important in responding to abiotic stimuli. It has been proposed to have great potential for improving the stress resistance of major agricultural crops. On the other hand, SA utilization is dependent on the dosage of the applied SA, the technique of application, and the status of the plants (e.g., developmental stage and acclimation). Here, we reviewed the impact of SA on saline stress responses and the associated molecular pathways, as well as recent studies toward understanding the hubs and crosstalk between SA-induced tolerances to biotic and saline stress. We propose that elucidating the mechanism of the SA-specific response to various stresses, as well as SA-induced rhizosphere-specific microbiome modeling, may provide more insights and support in coping with plant saline stress.

## 1. Introduction

The global food supply is under severe strain, with overall food consumption expected to increase by 70–85 percent to feed 9 billion people for the increased world population by 2050 [1]. However, abiotic challenges, such as high temperature, salinity, drought, and cold, are important factors that restrict agricultural productivity. As a result of the effect of human activities and the intensification of climate change, crops are subjected to increasing abiotic stressors, which have become the primary cause of agricultural system degradation and grain production decline [2,3]. Furthermore, the increased demand has compelled more crops to be grown on virgin land that is frequently subjected to abiotic stress due to insufficient irrigation, salinity, metal toxicity, and a lack of nutrients. One of the most severe constraints is saline stress caused by soil salinization [4]. Salinization is the process of raising the concentration of total dissolved salts in soil and water, which can occur naturally (primary salinization) or as a result of anthropogenic actions (secondary salinization). Climate change enhances the amount of salinization that occurs, through either a rise in sea levels or higher evaporation during droughts, while the use of saline groundwater and low-quality wastewater for irrigation exacerbates the secondary salinization process [4,5]. Saline stress has a direct adverse effect on plant growth, development, and yield due to its impacts on plant biochemistry and physiology. It extends beyond all stages of development, from seed germination to maturity. Under moderate salinity environments, all the important glycophytic crops reduce their average yield by 50–80% [5]. As a consequence, the strategy of sustainable development will include planting salt tolerance varieties, as well as inducing cultivated plants to resist or tolerate abiotic stresses through previously unknown resistance mechanisms. For instance, salicylic acid (SA) has been proposed to play an important role in resistance and defense signaling in both saline and biotic stress [6,7].

SA is an endogenous small-molecule phenolic compound that acts as a signal sensor to regulate plant response. It protects plant cells from the toxicity of ion accumulation and cell death by managing processes such as antioxidant defense, nitrogen metabolism, photosynthesis, and water stress [8,9]. The variations in SA levels among plant species, as well as due to environmental challenges, are quite dramatic [10]. The effect of endogenous SA levels has been related to a plant’s developmental stage and exogenous stimulation levels [11]. Furthermore, plant tolerance to salt, drought, heat, cold, and heavy metals can be induced by exogenous SA treatment [12,13,14,15,16]. Abiotic stress may induce serial gene expressions in plants, some of which are related to SA-dependent activation. These genes have an impact on a variety of biological processes, including molecular chaperones, antioxidant biosynthesis, and secondary metabolite synthesis [17,18,19].

Significant progress have been made in signal transduction under saline stress in recent years, particularly in the early signaling induced by salt. The significant role of both calcium waves and reactive oxygen species (ROS) and their downstream targets has been demonstrated, while the cell wall has frequently been implicated as a modulator of cell expansion during salt stress [20]. On the other hand, significant progress has been achieved with the biosynthetic pathway of SA in plants. Despite many studies having been conducted on the role of SA in plant salt stress, information on relatively in-depth mechanisms is still limited. Here, we propose to integrate the biosynthesis of SA, the role of SA in various abiotic stresses, and the potential mechanism of SA-mediated plant tolerance to salt stresses in this review. We also discuss the similarities and differences in SA signal transmission modes when plants are subjected to salt or biotic stresses. This information is beneficial for a better understanding of the role of the SA signal system in a stressed plant. Our understanding of signal pathways and growth adaptation mainly comes from the model plant *Arabidopsis*, which has not been directly transferred to crops. Plant saline tolerance is a complex trait that can manifest itself in a variety of ways, including ion accumulation, tissue-specific growth rate, biomass production, and seed production. Different salt reactions must be optimized for different crops in order to increase yield. The interplay of the SA signaling pathway and saline tolerance signaling will allow for the development of novel salt-tolerant crop varieties [4].

## 2. SA Biosynthesis in Plants

SA is initially synthesized from chorismate in plants through two independent pathways: the phenylpropane (PAL) and isochorismate synthase (ICS) pathways. In *Arabidopsis*, the ICS pathway plays the predominant role in the production of SA during plant defense [21]. Isochorismate synthase 1 (ICS1, also known as SID2 or EDS16) converts chorismate into isochorismate in the chloroplast [22]. *ics1* is sensitive to salinity [23], implying that SA biosynthesis through the ICS pathway is important for saline tolerance in plants. A chloroplast outer envelope protein named enhanced disease susceptibility 5 (EDS5) has also been characterized as a multidrug and toxin extrusion (MATE) transporter family protein; it transports the isochorismate from plastids to the cytosol [24,25]. In some bacteria, isochorismate is directly converted to SA by isochorismate pyruvate lyase (IPL), which has not been identified in any homologs in plants [26]. In plants, recent studies revealed that isochorismate could be converted to SA by AvrPphB Susceptible 3 (PBS3) in the cytosol via a completely different mechanism than that in bacteria [25,27]. PBS3 serves as a GH3 acyl-adenylate/thioester-forming enzyme, which catalyzes the conjugation of isochorismate to glutamate to produce a biosynthetic intermediate isochorismate-9-glutamate (IC-9-Glu). IC-9-Glu can spontaneously decay into SA [25] or be converted to SA by enhanced pseudomonas susceptibility 1 (EPS1) [27].

Using PAL defective mutants or treatment with specific inhibitors, the PAL pathway was also identified with SA production, independent of the ICS pathway, in *Arabidopsis* [28]. Isotope tracing experiments in tobacco were performed to identify the complete steps for SA production in the PAL pathway. Trans-cinnamic acid (*t*-CA) was considered to be first synthesized from phenylalanine (Phe) and then converted to SA via benzoic acid (BA) [29]. The conversion of Phe to *t*-CA by PAL is one of the rate-determining steps in SA biosynthesis. The means by which the PAL pathway promotes SA synthesis is still controversial. A recent study indicated that the PAL gene catalyzes phenylalanine to finally synthesize 4-hydroxybenzoic acid (4-HBA) rather than SA [30]. In any case, it is clear that BA is the precursor of SA. Abnormal Inflorescence Meristem 1 (AIM1), encoding a 3-Hydroxyacyl-CoA dehydrogenase involved in β-oxidation, is required for BA production in rice roots [31]. The means by which the enzymes determine the final conversion of BA to SA remains unclear; some studies suggest that benzoic acid 2-hydroxylase (BA2H) or the cytochrome 450 family protein may play a role [32]. Salinity increases rice endogenous SA levels by increasing BA2H activity, suggesting that the PAL pathway is involved in saline stress [33].

The SA synthesis pathway may vary in other crops. Rice accumulates high levels of SA, approximately 10-fold more than that of *Arabidopsis* leaves under normal growth conditions [10]. The *OsPAL6*-knockout mutant showed about a 60% decrease in the basal SA level and increased susceptibility to the fungal pathogen *Magnaporthe oryzae* [34]. The SA levels were also significantly reduced in *osaim1* [31]. Interestingly, both the ICS and PAL pathways were shown to contribute to pathogen-induced SA accumulation in soybeans. Silencing of either the *ICS* or *PAL* genes led to a severe reduction in SA accumulation during infection by the bacterial and oomycete pathogen [35]. These findings indicate that significant variations exist in different plant species for the contributions of the two SA biosynthesis pathways. Thus far, no clear orthologs of PBS3 or EPS1 have been identified in most plant species besides the Brassicaceae family; the role of the ICS pathway for SA synthesis in the plant kingdom needs more evaluation [36]. Similarly, *PAL* genes are also involved in the production of many secondary metabolites that are unrelated to SA, such as flavonoids. In order to better understand the function of the PAL pathway, it is necessary to identify the BA2H enzymes involved in the conversion of BA to SA in the future.

SA biosynthesis is strictly under regulation during biotic and abiotic stress. As summarized in Figure 1, TCP (Teosinte-like1/Cyclidea/PCF), WRKY (WRKY DNA-binding protein), and CBP60 (CAM Binding Protein 60) proteins have been identified as key transcription factors that control ICS1 expression [9,37]. Among these, the CBP60 proteins of SARD1 (systemic acquired resistance (SAR) defect 1) and CBP60g have been extensively studied [38]. Their expression was induced via TGA1 (TGACG-binding factor 1) and TGA4 by pathogen inoculation, which further binds to the promoter regions of ICS1, EDS5, and PBS3 to activate the SA synthesis-related gene expression [39,40]. CBP60b could influence the expression of *SARD1* and *CBP60g* to modulate SA accumulation [41,42]. Coronatine stimulates the ANAC (abscisic acid-responsive NAC) transcription factors ANAC019, ANAC055, and ANAC072 via MYC2 to suppress the expression of *ICS1* and lower the synthesis of SA [43]. Other transcription factors that suppress *ICS1* expression include WRKY54, WRKY70, EIN3 (Ethylene Insensitive 3), and CBP60a [9,44]. Meanwhile, the atypical E2F transcriptional repressor DEL1 could suppress *EDS5* expression [45].

## 3. Signal Transduction Pathway of SA

The produced SA in plants needs to be perceived by its receptors in order to trigger the downstream response. Several SA-binding proteins (SABPs) have been identified to perceive SA. NPR1 (non-expresser of PR genes 1), an SA-binding protein, has been well studied to play a crucial role in activating downstream disease resistance genes [46,47]. Pathogens promote phosphorylated NPR1, which is recruited to cullin 3 ubiquitin ligase for a series of physiological and biochemical reactions. The phosphorylated NPR1 is essential for SAR induction [36,48]. The monomer NPR1 is translocated into the nucleus for phosphorylation and interacts with bZIP (basic leucine zipper protein) transcription factor TGA family proteins to activate the expression of downstream resistance-related genes [9]. A recent study revealed that SA activates NPR1 in the nucleus to induce SAR and enhances the formation of NPR1 condensates during plant immunological responses [49]. Meanwhile, NPR1-dependent SA signaling is also central to saline and oxidative stress tolerance in *Arabidopsis* [50].

In addition, other NPRs operate as SA receptors and play a key regulatory role in the SA signal transduction cascade in *Arabidopsis*. NPR3 and NPR4 can function as E3 ubiquitin ligase linkers to mediate NPR1 degradation, during which SA sensing systems influence cell death and survival [51,52]. NPR1, NPR3, and NPR4 bind SA with varying affinities and play a role in binding to downstream TGA transcription factors in response to pathogen infections [53]. Moreover, the NPR3/4-mediated repression of *CBP60g* and *SARD1* expression occurs under drought and bacterial combined stress in an SA-dependent manner, showing that NPR3/NPR4 is involved in plant resistance to abiotic stress [54]. NPR3/4 combine with TGA to inhibit downstream gene expression, whereas SA can relieve this inhibition [55]. As a result, two parallel signaling pathways may exist downstream of SA. On the one hand, when the concentration of SA in plants is very low, NPR3/4 block the expression of genes downstream of SA; the transcriptional repression effect of NPR3/4 on SA downstream genes is relieved based on elevated SA concentrations during pathogen infection. On the other hand, the accumulation of SA in plants activates the transcriptional activation activity of NPR1, which in turn induces the expression of downstream genes of SA [56].

For SA signal transduction, EDS1 (enhanced disease susceptibility 1) and PAD4 (phytoalexin deficiency 4) were regarded as essential upstream components of SA-dependent and SA-independent basal defense pathways [57]. They belong to the lipase-like enzymes family. The N-terminus of EDS1 and PAD4 interact with each other to form a binary complex. Broken interactions between EDS1 and PAD4 can hinder the induction and synthesis of SA and weaken the resistance to pathogenic bacteria [58]. SA accumulation will upregulate the expression of EDS1 and PAD4, thus forming a positive feedback signal loop [59]. EDS1 was protected by PBS3 from proteasome-mediated degradation, and the PBS3–EDS1 relationship may also boost EDS1 activity in SA and basal immunity [60].

## 4. Physiological Responses by SA during Saline Stress in Plants

The world’s saline soil area is close to 1 billion hm^2^, accounting for approximately 7% of the earth’s land area. It is extensively spread across Asia, the Americas, Australia, Africa, and other regions [61]. Furthermore, soil salinization threatens 77 million hm^2^ of land, and salinization has lowered productivity on roughly one-third of irrigated agriculture [62,63]. Meanwhile, as a consequence of global warming, the salinization area is dramatically increasing on a daily basis [4,64]. Salinization produces osmotic stress as well as ionic toxicity in cells. Plant production losses are frequently caused by an imbalance in cellular ionic and osmotic balances [4]. Furthermore, salt can cause plant stomatal closure and reduce intercellular CO_2_ concentration in the early stages of stress; in the latter stages, the chloroplasts become damaged and chlorophyll synthesis is blocked, affecting photosynthesis and resulting in decreased biomass, slowed plant growth, and even death [65,66,67]. SA can regulate important plant physiological processes, including photosynthesis, nitrogen metabolism, antioxidant defense system control, and water use efficiency improvement [8]. The pretreatment of rice with SA during germination during saline stress was shown to dramatically enhance the shoot and root lengths, resulting in greater salinity resistance [68]. SA was found to increase photosynthesis and the antioxidant response to boost yield components in the ASD16 and BR26 rice lines [69]. The participation of SA in ion transport, blooming, and photosynthesis may account for the contributions to the yield [70].

Saline stress not only affects the ability of plants’ roots to absorb water but also destroys leaf cells through transpiration, reducing plant growth. A Na^+^ overabundance in the soil will disrupt the ion dynamic equilibrium, causing a variety of stress responses, such as osmotic stress and ion stress in plants [18]. Excess Na^+^ ions also cause a K^+^ deficit and hinder a range of K^+^-dependent biological processes in plants. As a result, one of the primary symbols for measuring plant salt tolerance is the dynamic balance of Na^+^ and K^+^ in plant cells [65,70]. Saline stress also limits the absorption of other ions, such as Ca2^+^ and Mg2^+^, resulting in a nutritional shortfall [4]. SA may lower the Na^+^ and Cl^−^ levels in the cell. Exogenous SA treatment was found to elevate internal SA levels and raise the K^+^ content, which decreased the Na^+^ levels in *Arabidopsis* and mung bean under saline stress, causing physiological changes in the plants [71].

A substantial number of reactive oxygen species (ROS) accumulate in plants in high-salt environments, reducing the fluidity of the cell membrane, increasing the permeability of the cell membrane, causing metabolic instability of the membrane system, and finally resulting in oxidative damage [72,73]. Some enzymes involved in oxidative stress, such as superoxide dismutase (SOD), catalase (CAT), peroxidase (POX), and glutathione reductase (GR), elevated their enzymatic activities during saline stress [74]. However, these protein levels decreased within the cells and, therefore, restored the physiological mechanisms in plants after SA treatment [70,75]. Consistent with the physiological response, exogenously SA-treated plants showed increased expression in the transcripts of various antioxidant components, such as dehydroascorbate reductase (DHAR), glutathione peroxidase (GPX1, 2), glutathione synthetase (GS), and glutathione S-transferase (GST1, 2), under saline stress [76]. In the case of high saline stress, oxidative stress was minimized by the SA-mediated reduction of cellular malondialdehyde (MDA) and ROS production in *Hordeum vulgare* [8].

In addition, the buildup of osmotic regulators can improve plant saline tolerance. Plants resist salt-induced osmotic stress by maintaining the osmotic pressure equilibrium within and outside the cell, thereby managing the accumulation of organic or inorganic osmotic regulators in the cytoplasm [64,66,77]. Cl^−^ and K^+^ are two major inorganic osmoregulators in plants; they do not require synthesis through metabolism [78]. Alanine, glutamate, asparagine, glycine, betaine, and sucrose are examples of organic osmoregulatory chemicals; the synthesis of these compounds increases considerably when exposed to saline stress. Betaine is an organic osmoregulation substance involved in the response to a variety of abiotic stressors. Plants can boost betaine synthesis in response to high salt, high temperature, and drought [64,79]. Treatment with 0.5 mM SA maximally increased the accumulation of glycine betaine (GB) (>40%), which acted against 50 mM NaCl-accrued impacts by minimizing the accumulation of Na^+^ and Cl^−^ ions and, hence, oxidative stress [80]. Sugars and free amino acids are also key organic osmoregulators, and their buildup in the cytoplasm can improve saline tolerance in plants [81,82]. In particular, the concentration of proline has been utilized as an important physiological marker to quantify stress resistance in plants [83]. During exogenously applied SA, proline metabolism became significantly altered, leading to the maintenance of turgor by the accumulation of higher levels of free proline in lentils to enhance their saline tolerance [84].

## 5. Molecular Mechanisms Controlled by SA Regulation in Response to Saline Stress

The role of SA in strengthening salinity stress tolerance mechanisms has been reported in cereal plants such as rice, wheat, barley, and maize [85,86,87,88], and in other plant species including *Cucumis sativus* [89], *Solanum lycopersicum* [90], *Vicia faba* [91], and *Medicago sativa* [92]. Some SA synthesis genes, such as *OsICS* and *OsPAL*, show induced expression under saline stress [18,88]. Meanwhile, the introduction of the bacterial salicylate hydroxylase gene (NahG) into plants resulted in stronger tolerance to high saline stress than that of the wild type [93]. 

Under saline stress, the water potential of the soil solution is reduced, resulting in osmotic stress [94]. Plants have osmotic stress sensors to monitor osmotic stress and convert it into Ca2^+^ signals. Ca2^+^ signals are interpreted by Ca2^+^ binding proteins, which govern the plant’s adaptive responses to osmotic stress [70,95,96]. OSCA1 (Reduced hyperosmolality induced [Ca2^+^]i increase 1) is a Ca2^+^ channel that is regulated by high osmotic pressure in *Arabidopsis*. It selectively responds to exogenous osmotic stress as a receptor [95]. In addition, maintaining Na^+^ homeostasis at the cellular level is a crucial aspect of plant saline tolerance [77,97]. Na^+^ in soil mostly enters plant root cells through non-selective cation channels or high-affinity K^+^ transporters [97]. Significant progress has been made in the identification of MOCA1 (monocation-induced [Ca2^+^]i increases 1), which is expected to operate in extracellular salt sensing, including but not limited to Na^+^ ions [98]. The early reaction to Ca2^+^ waves that occurs in response to Na^+^, K^+^, or Li^+^ ions is absent in the *moca1* mutant. In conclusion, Ca2^+^ should be the downstream signal molecule after saline stress perception. Ca2^+^ is also closely related to plant immunity and SA signaling. For example, a Ca2^+^/calmodulin-binding transcription factor AtSR1 directly interacted with the promoter and regulated the expression of EDS1 to modulate the SA content [99].

Several critical components that regulate Na^+^ homeostasis have been identified in plants. The function of the HKT1 (high-affinity K^+^ transporter) family of Na^+^ transporters is mainly to mediate the unloading of Na^+^ in xylem vessels, reduce the Na^+^ transport from the roots to shoots, lower the Na^+^ content in the shoots, and increase the saline tolerance [100]. NPR1-mediated SA signaling has been argued to be pivotal for controlling Na^+^ entry into the roots and the subsequent long-distance transport into the shoots, enhancing the H^+^-ATPase activity in the roots and increasing the K^+^ concentration in the shoots during saline stress [50]. The same study also reported that high endogenous-SA mutant *nudt7* decreased the shoot Na^+^ concentration during prolonged saline stress [50]. 

The Na^+^/H^+^ reverse transporter NHXs (Na^+^/H^+^ exchanger 1), which are found on the vacuole membrane, store excess Na^+^ in the vacuole through ion regionalization, lowering the Na^+^ levels in the cytoplasm [101]. AtSOS1 is the principal Na^+^/H^+^ reverse transporter in *Arabidopsis*, and its transport activity is controlled by the proton transmembrane gradient maintained by plasma membrane H^+^-ATPase. Saline tolerance can be considerably increased by the overexpression of AtSOS1 [102]. In the tomato plant, NHX1 and SOS1 transcripts were found to be significantly altered in SA-treated plants; SA regulated the ion transporter transcription and maintained ion homeostasis [103].

When saline stress occurs, the intracellular free Ca2^+^ concentration increases instantly. The Ca2^+^ signal is detected by the Ca2^+^ binding protein SOS3, which stimulates the association of SOS3 with serine/threonine protein kinase SOS2. Then, the activated SOS2 can phosphorylate and activate SOS1 to promote the maintenance of Na^+^ homeostasis and saline tolerance [4]. The function of the SOS pathway has been described in numerous species, such as rice and barley, implying that the SOS system-mediated regulation of Na^+^ homeostasis is conserved [66]. These processes work together to improve plant saline tolerance by regulating Na^+^ homeostasis at the tissue and cellular levels [77].

Potassium (K) is an important mineral for plants. It is intertwined with a variety of vital daily activities. The content of K^+^ in the cytoplasm of *Arabidopsis* roots reduced quickly after they were exposed to salt. A plant’s saline tolerance can be considerably improved by increasing the intracellular K^+^ levels [103]. K^+^ uptake and efflux are mediated by K^+^ transporters (such as *AKT1*, *Arabidopsis* K^+^ transporter 1) or *KORC* (potassium outward rectifying channels) in plants. These genes are regulated by the transmembrane potential gradient and play a role in the steady-state maintenance of intracellular K^+^. Saline stress reduces the intracellular K^+^ concentration by destroying the transmembrane potential, causing membrane depolarization and inhibiting K^+^ transporter activity [104]. Ion transporters of the HAK (High-affinity K^+^)/KUP (K^+^ uptake permease)/KT family were initially discovered through high-affinity potassium transport abilities in fungi. They are involved in root K^+^ absorption and long-distance K^+^ transport from root to shoot in *Arabidopsis* and rice [105]. *AtKUP*, *OsHAK1*, *OsHAK5*, and *OsAKT1* govern the long-distance transport of K^+^ by participating in K^+^ loading in the xylem, and then altering the K^+^ levels in shoots and the plant saline tolerance in rice or *Arabidopsis* [106]. Interestingly, they are also involved in the regulation of Na^+^ unloading in the xylem. For instance, *ZmHAK4* may limit Na^+^ transport from the roots to shoots by moving Na^+^ in the xylem to the adjacent parenchyma cells, hence increasing salt resistance in maize [107]. SA can restore the membrane potential and limit salt-induced potassium efflux via the GORK (GATED OUTWARDLY-RECTIFYING K+ CHANNEL) channel, resulting in saline tolerance [23,71]. SA priming can be an important strategy for enhancing major GSH-based H_2_O_2_-metabolizing enzymes, such as GST. To this end, SA mitigated salinity stress injury in *Solanum lycopersicum* by causing characteristic changes in the expression pattern of *GST*- family members, such as *SlGSTT2*, *SlGSTL3*, and *SlGSTF4* [108]. In another example, SA can regulate the generation of NaCl-induced oxygen and nitrogen reactive species in rice to induce the upregulation of *OsGSNOR* and increase the SNO content, resulting in considerable biomass accumulation under stress conditions [109,110].

## 6. Mechanism Comparison of SA in Saline and Biotic Stress

As a signal molecule, SA governs plant responses to both biotic and abiotic stress via a complex signal transduction network. In the molecular signaling pathway, plants share several key components, including Ca2^+^-driven and ROS-driven signals, among saline and pathogen infections. These two signals are regarded as the upstream of the SA pathway. However, there are differences in their specific mechanisms. For signal perception, studies have revealed that Ca2^+^ signal channels regulate the immune process, such as CNGC2, CNGC4, and OSCA1.3 [111,112], triggering a cascade of downstream immunological responses to connect with the SA pathway. Previously, glycosyl inositol phosphorylceramide (GIPC) sphingolipids generated by MOCA1 were shown to mix with monovalent cations and cause Ca2^+^ influx to respond to salinization; however, the Ca2^+^ channel that is involved has yet to be identified [98]. It is vital to note that the calcium channels that plants employ to detect biological stress and saline stress are distinctive. 

Some of the described responses depend on cytosolic Ca2^+^ concentrations [9]. SOS3/CBL4 perhaps function as the signal hub of these two stresses. Ca2^+^ is detected by SOS3/CBL4 in conjunction with SOS2/CIPK24. This mechanism is supplemented by CBL10, which can form a complex with SOS2, which is involved in Na^+^ chelation in vacuoles [113]. In addition, the SOS3/CBL4-CIPK6 complex could regulate K^+^ allocation, and the *cipk6* mutant was found to enhance disease resistance with increased SA accumulation [114]. Normally, several calcineurin B-like proteins (CBL) bind Ca2^+^ and promote protein phosphorylation via contact with CBL interacting protein kinase (CIPK) to regulate the output of Ca2^+^ flow, thereby adjusting the salt or immune response, respectively [115,116], such as CPK3, which phosphorylates the vacuolar K^+^ channel TPK1 (two pore K+ channel 1). In agreement with that, both *cpk3* and *tpk1* mutants were sensitive to salt [117]. The CBL2/CBL3-CIPK3/CIPK9/CIPK23/CIPK26 pathways enhance salt tolerance by regulating ion homeostasis, such as Ca2^+^ and K^+^ [114]. On the other hand, some CBL-CIPK modules act as regulator of pathogen-mediated programmed cell death (PCD) and SA transduction. SlCBL10-SlCIPK6 and TaCBL4-TaCIPK5 participate in plant immunity through ROS content alteration and downstream SA signals in tomato and wheat, respectively [118,119]. Enhanced resistance in *Arabidopsis cipk14* mutants was also observed, which could be explained by a rise in SA accumulation and the expression of defense-related genes (*PR1*, *EDS1*, *EDS5*, *ICS1*) [120]. Parts of CBLs or CIPKs still need to have their biological functions clarified, particularly when compared functionally under saline or biological stress. This information could fill in the knowledge gaps in the SA signaling network.

Both saline stress and biotic stresses rapidly produce elevated levels of extracellular ROS molecules, such as hydrogen peroxide, singlet oxygen, superoxide, and hydroxyl radicals, which destroy redox homeostasis and cause oxidative damage to plant cells [121]. Both ROS and SA were found to be involved in defense signaling and the regulation of cell death. Some PAMP molecules trigger immunity (PTI) through phosphorylated RBOHD to induce reactive oxygen species (ROS) production [122], and H_2_O_2_-mediated SA accumulation could in part be explained by the catalytic activity of H_2_O_2_ on the BA2H enzyme involved in the conversion of benzoic acid to SA [31,123]. In *Arabidopsis*, the expression patterns of RBOH genes change dynamically. They produce ROS waves under saline stress within 24 h [124], indicating that the complex ROS production network is continuously active and plays a central role in the early salt response; the subsequently produced SA may be involved in the oxidative stress response of plants through its antioxidant activity (Figure 2).

The fundamental mechanism by which SA lessens salt damage may be its capacity to prevent K^+^ efflux through GORK [71]. To preserve the beneficial link between K^+^ retention in plant roots and salt tolerance, SA can reduce the effects of salinity and increase the K^+^ concentration in the roots [104]. The plasma membrane proton pump (H^+^-ATPase) activity, which plays a crucial role in regulating GORK function, is also lacking in the *npr1* mutant, suggesting that SA-increased H^+^-ATPase activity is NPR1-dependent [50]. NPR1, the primary SA receptor protein, is important for many physiological processes as well as different types of plant disease resistance. It is interesting to note that saline stress causes rapid accumulation of the NPR1 protein in the chloroplast stroma, which prevents stress from causing a reduction in the photosynthetic capacity in a redox-dependent manner [125,126]. This suggests that the NPR1 protein performs an unidentified function in the response to saline stress.

Several transporters contribute to Na^+^ absorption and efflux during saline stress (HKT, AKT1, GOAK, OSCA1, etc.). The membrane depolarizes when Na^+^ enters the cell. Since many K^+^ and Ca2^+^ channels are voltage-regulated, this shift in membrane voltage affects ion transport. H^+^-ATPase action can repolarize the membrane. Na^+^ binding to Glycosyl-Inositol-Phosphoryl-Ceramides (GIPCs) in the plasma membrane is required for the formation of saline stress-induced Ca2^+^ signals. Extracellular Na^+^ directly binds to negatively charged GIPCs (catalytic production by MOCA1) in the plasma membrane’s outer leaflet under saline stress. Na^+^-bound GIPCs may directly activate/regulate Ca2^+^ channels. The Ca2^+^ sensor protein CBL4 (SOS3) decodes the saline stress-induced Ca2^+^ signal by activating CIPK24 (SOS2), which then phosphorylates and activates SOS1 to pump excess Na^+^ out of the cell. The osmotic and saline stress-sensing components, OSCA1 and GIPCs, have a direct influence on the formation of the major osmotic/salt stress-induced Ca2^+^ signal; however, this is questionable in the case of AKT1 (K^+^). For AKT1, a regulatory function for Ca2^+^-dependent phosphorylation via CBL1/9-CIPK23 has been identified. Meanwhile, for OSCA channels, kinase-dependent regulation has been characterized; however, no evidence of a direct relationship with Ca2^+^ has yet been revealed. The plant hormone ABA can similarly influence SA synthesis by promoting the production of Ca2^+^ and ROS. Simultaneously, SA feedback influences ABA. SA increases the activity of H^+^-ATPases via NPR1 and decreases K^+^ leakage via the GORK channel. SA also modulates other salt response responses via unknown mechanisms (probably through NPR1).

## 7. Summary and Future Prospects

SA is a hormone that plays a role in a plant’s reaction to biological stress. Plant response to saline stress has been linked to a variety of genes involved in protein synthesis and signal transmission, including SA-related genes such as *ICS1*, *EDS5*, *PAL*, and *NPR1* [18,125]. However, it is unclear whether these reactions are triggered by the SA signal feedback control or the gene’s function. Saline stress response has also been linked to transcription factors downstream of SA, such as TGAs; however, the molecular mechanism remains unclear [127]. In fact, in trying to improve plant resistance to biological stress or saline stress, applying different amounts of SA to various plant species will result in contrasting outcomes. This could be due to the various treatment methods used, such as foliar spray or hydroponic culture. However, the response mechanism to SA may differ between plant species. In general, low concentrations or brief applications of SA can improve plant tolerance to abiotic stress, whereas high concentrations or the continuous application of SA can inhibit plant growth and reduce stress tolerance [128]. In any case, the application of SA holds a lot of promise.

The signaling mechanism of SA under biotic stress is generally obvious. It is unknown how SA plays a specialized function in saline stress and how each stress differentiates the accumulation of SA. If we can mechanistically analyze how plants discriminate the induction of SA by each stress and identify the promoters specifically induced by salt stress in the SA synthesis pathway, we will be able to better understand the specificity of plant responses to biotic or saline stress and apply SA more effectively.

Unlike drought, cold, heat, and other systemic stresses, a plant’s sensing of salt is entirely dependent on plant roots. With the rapid advancement of microbiome research, we should emphasize that plant saline tolerance is not only derived from the plant genome but also from symbiotic microbes, which may have a substantial impact. Rhizosphere bacteria can encourage plants to create systemic tolerance, regulate plant physiological responses, and ensure proper growth in high-salt environments [129]. Stress may also affect plant physiology and immunology, as well as soil parameters (PH, O_2_, and nutrient levels), by changing the content and concentration of root exudates and increasing the enrichment of certain microbes in the rhizosphere [130]. Many rhizosphere bacteria can produce SA [131] to control the microbial colonization of certain bacterial groups in the roots. In *Arabidopsis* root flora, there are considerable variations between SA synthesis and wild-type mutants; however, it is unknown how SA production attracts some bacteria while blocking others. Some selective microbes may employ SA as their carbon source [132]. Understanding the relevant mechanisms regarding how SA promotes the formation of certain microbial communities in the plant rhizosphere to promote saline stress tolerance is also a favorite topic for future study.

## Figures and Tables

**Figure 1 ijms-24-03388-f001:**
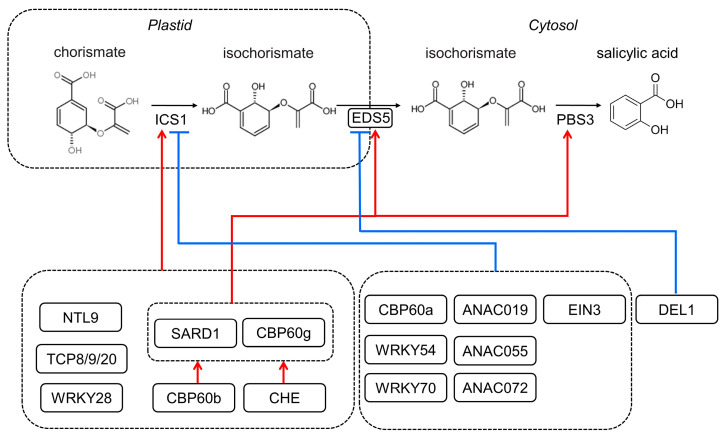
The simplified transcriptional regulatory network for SA biosynthesis in *Arabidopsis*. The enzymes required for SA synthesis via the isochorismate (IC) pathway are encoded by the genes *ICS1*, *EDS5*, and *PBS3*. Transcription factors, including NTL9, TCP8/9/20, WRKY28, SARD1, and CBP60g, may positively regulate *ICS1* gene expression; SARD1 and CBP60g also activate *EDS5* and *PBS3* gene expression (red arrow); and CBP60b and CHE control downstream genes by modulating SARD1 and CBP60g. TCP8/9 interact with WRKY28 and SARD1, which may play a role in their transcriptional control. CBP60a, WRKY54/70, ANAC019/055/072, and EIN3 negatively regulate *ICS1* gene expression (blue line), while DEL1 suppresses *EDS5* gene expression.

**Figure 2 ijms-24-03388-f002:**
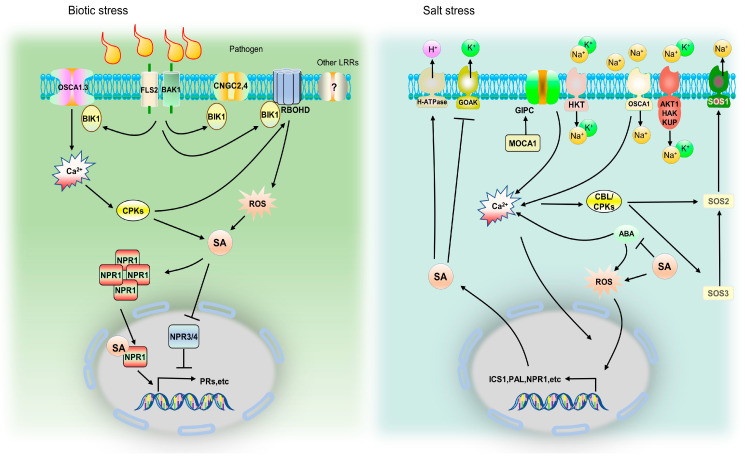
The signaling pathway of salicylic acid mediated by pathogens and saline stress. Pathogen-associated molecular patterns (PAMPs) are recognized by some receptor-like kinases (RLKs, such as FLS2 and BAK1) and activate PAMP-triggered immunity (PTI). The binding of flg22 triggers the dissociation of BIK1 from the FLS2-BAK1 complex. Then, BIK1 phosphorylates RBOHD to induce reactive oxygen species (ROS) production. BIK1 also phosphorylates and activates the CaM-gated OSCA1.3 and CNGC2-CNGC4 channels, triggering Ca2^+^ to enter the cell. Ca2^+^ binding to EF-hand motifs in RBOHD and CDPKs activates RBOHD and boosts ROS production, hence propagating the defense signal and activating SA biosynthesis. In the SA signaling cascade, NONEXPRESSER OF PR GENES 1 (NPR1) monomerization requires SA binding to NPR1 as well as redox regulation and leads to the expression of pathogenesis-related (PR) genes via NPR1-TGA transcription factor interactions. NPR3 and NPR4 proteins (paralogs of NPR1) also bind SA while acting as transcriptional corepressors.

## Data Availability

Not applicable.

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
