# Peer review of "Emerging Roles of Salicylic Acid in Plant Saline Stress Tolerance"

_ijms, 2023, doi:10.3390/ijms24043388_

Round 1

Reviewer 1 Report

It is interesting to understand the crosstalk between SA-induced tolerances to biotic and saline stress. The author reviewed the impact of SA on pathogens and saline stresses. This will provide more insights and support in coping with plant saline stress. Some minor suggestions:

1. As mentioned in the text “significant variations exist in different plant species for the contributions of the two SA biosynthesis pathways” and “no clear orthologs of PBS3 or EPS1 have been identified in most of plant species beside the Brassicaceae family”. Figure 1 is suggested to named as “The simplified transcriptional regulatory network for SA biosynthesis in Arabidopsis”.

2. Please improve the resolution of Figure 2.

Author Response

Thanks for your comments. we have corrected the manuscript according to each of your suggestions.

  1. As mentioned in the text “significant variations exist in different plant species for the contributions of the two SA biosynthesis pathways” and “no clear orthologs of PBS3 or EPS1 have been identified in most of plant species beside the Brassicaceae family”. Figure 1 is suggested to named as “The simplified transcriptional regulatory network for SA biosynthesis in Arabidopsis”.

Response: Thanks for your good suggestions. The corresponding description have added into the revised manuscript.

  1. Please improve the resolution of Figure 2.

Response: Thanks for your suggestion. We have changed to high resolution pictures in the revised version.

Reviewer 2 Report

The paper describes the role of salicylic acid in the salt stress tolerance of plants. The article is well-organized and adequately divided into sections and is well-illustrated.

The authors have thoroughly reviewed the literature bringing in biochemical and molecular mechanisms involved in the role of salicylic acid promoting salt tolerance in plants. For the information of the authors, an important paper on the theme (Riva-San Vicente & Plasencia, 2011. Salicylic acid beyond defense: Its role in plant growth and development. Journal of  Experimental Botany, 62(10): 3321-3338.)  is missing, the same can be consulted and if authors think it appropriate, it can be included. 

 A thorough English revision is recommended.

Author Response

Thanks for your comments. We have corrected each of your suggestions in the revised version. 

The authors have thoroughly reviewed the literature bringing in biochemical and molecular mechanisms involved in the role of salicylic acid promoting salt tolerance in plants. For the information of the authors, an important paper on the theme (Riva-San Vicente & Plasencia, 2011. Salicylic acid beyond defense: Its role in plant growth and development. Journal of  Experimental Botany, 62(10): 3321-3338.)  is missing, the same can be consulted and if authors think it appropriate, it can be included. 

 A thorough English revision is recommended.

Response: Thanks for your suggestion. This reference paper is important and suitable for our manuscript, we have added it to the proper section of the text. Also, we have asked for language editing from the recommendation author service from the MDPI website.

Reviewer 3 Report

The manuscript provides a point of view of the role of SA in saline stress alleviation by plants. Therefore it provides a good perspective for researchers working in the field and would be of interest to be published in the International Journal of Molecular Sciences. In order to do so, major revision needs to be undertaken by authors.

Linguistic and/or grammatical addressing is needed everywhere throughout the manuscript. Some non-exclusive examples are listed: “Salicylic acid (SA) is a key phytohormone of phenolic chemical compound”; “The role of SA has received a lot of attention on plant-pathogen interactions”; “SA also play vital role”; “great potential to broadly improve stresses resistance”; “SA-induced tolerances to biotic and saline stress”; “virgin land that frequently subjected to abiotic stress”; “plant tolerant to salt, drought, heat, cold, and heavy metals can be induced by exogenous SA treatment”; “Significant progress had been made in signal transduction under saline stress in recent years”; “On the other hand, the biosynthetic pathway of SA in plants has achieved significant progress.”; “The generated SA need to be perceived by its receptors and triggered the downstream response.”; “The over accumulation of Na+ in the soil will disrupt the ion dynamic equilibrium” ,and so on everywhere through the text.

The title and closing sentence of the Abstract imply that addressing saline stress is the main target of SA research in the review paper. If so, I would suggest adding some more information in the Introduction of factors leading to soils becoming saline worldwide.

Special caution is needed in enlisting protein/enzymes/genes names. For example as it is written now, it seems as the EPS1 has been designated with the term “Pseudomonas susceptibility 1” (Line 89); “SAR” has been mentioned without explaining the abbreviation (Line 152), and so on.

The scientific content of the cited references has to be more precisely explained, for example “Saline stress not only affects plant roots' ability to absorb water, but also destroys leaf cells through transpiration, reducing plant growth [71].” is not very informative.

Author Response

Thanks for your suggestions and concerns. We have corrected each of them in revised version.

Linguistic and/or grammatical addressing is needed everywhere throughout the manuscript. Some non-exclusive examples are listed: “Salicylic acid (SA) is a key phytohormone of phenolic chemical compound”; “The role of SA has received a lot of attention on plant-pathogen interactions”; “SA also play vital role”; “great potential to broadly improve stresses resistance”; “SA-induced tolerances to biotic and saline stress”; “virgin land that frequently subjected to abiotic stress”; “plant tolerant to salt, drought, heat, cold, and heavy metals can be induced by exogenous SA treatment”; “Significant progress had been made in signal transduction under saline stress in recent years”; “On the other hand, the biosynthetic pathway of SA in plants has achieved significant progress.”; “The generated SA need to be perceived by its receptors and triggered the downstream response.”; “The over accumulation of Na+ in the soil will disrupt the ion dynamic equilibrium” ,and so on everywhere through the text.

Response: Thanks for your suggestions. We have corrected each of them according to your comments. Also, we have corrected the linguistic and grammar errors after asking for professional language service.

The title and closing sentence of the Abstract imply that addressing saline stress is the main target of SA research in the review paper. If so, I would suggest adding some more information in the Introduction of factors leading to soils becoming saline worldwide.

Response: Thanks for your suggestion. More information about the salinization have added into Introduction section.

Special caution is needed in enlisting protein/enzymes/genes names. For example as it is written now, it seems as the EPS1 has been designated with the term “Pseudomonas susceptibility 1” (Line 89); “SAR” has been mentioned without explaining the abbreviation (Line 152), and so on.

Response: Thanks for your good suggestion. We checked carefully throughout the manuscript about the gene names or abbreviation. In fact, we have explained and abbreviated SAR when referring to gene SARD (Line 125 of the original version or line 132 of the revised version).

The scientific content of the cited references has to be more precisely explained, for example “Saline stress not only affects plant roots' ability to absorb water, but also destroys leaf cells through transpiration, reducing plant growth [71].” is not very informative.

Response: Thanks for your suggestion. We checked the references to make the citations more accurate. 

Round 2

Reviewer 3 Report

The authors have considerably improved the manuscript. There is still need of some minor addressing with examples given below:

 Line 61: “had” needs to be replaced by “have”; Line 66: there needs to stand a full phrase, for example correct to: “elucidation of the biosynthetic pathway of SA”; Line 86: “ics1” the sentence needs to start with a capital letter; Line 209: “may dramatically enhance” has to be substituted with: “was shown to dramatically enhance”

Elaboration of scientific terminology is still needed. For example, please reconsider the term and its abbreviation: “isochorismate acid (ICS)”. The widely accepted term is “isochorismate synthase (ICS) pathway”.